# Reducing uncertainties in satellite estimates of aerosol-cloud interactions over the subtropical ocean by integrating vertically resolved aerosol observations

David Painemal[1,2], Fu-Lung Chang[1,2], Richard Ferrare[2], Sharon Burton[2], Zhujun Li[1,2], William L. Smith Jr. [2], and Patrick Minnis[1,2], Yan Feng[3], and Marian Clayton[1,2]

[1]Science Systems and Applications Inc., Hampton, Virginia, 23666, U.S.
[2]NASA Langley Research Center, Hampton, Virginia, 23691 U.S.
[3]Argonne National Laboratory, Lemont, Illinois, 60439, U.S.

*Correspondence to*: David Painemal (david.painemal@nasa.gov)

**Abstract.** Satellite quantification of aerosol effects on clouds relies on aerosol optical depth (AOD) as a proxy for aerosol concentration or cloud condensation nuclei (CCN). However, the lack of error characterization of satellite-based results hampers their use for the evaluation and improvement of global climate models. We show that the use of AOD for assessing aerosol-cloud interactions (ACI) is inadequate over vast oceanic areas in the subtropics. Instead, we postulate that a more physical approach that consists of matching vertically resolved aerosol data from the Cloud-Aerosol Lidar and Infrared Pathfinder Satellite Observations (CALIPSO) satellite at the cloud-layer height with Aqua Moderate-resolution Imaging Spectroradiometer (MODIS) cloud retrievals reduces uncertainties in satellite-based ACI estimates. Combined aerosol extinction coefficients ($\sigma$) below cloud-top ($\sigma_{BC}$) from the Cloud-Aerosol Lidar with Orthogonal Polarization (CALIOP) and cloud droplet number concentrations ($N_d$) from Aqua-MODIS yield high correlations across a broad range of $\sigma_{BC}$ values, with $\sigma_{BC}$ quartile correlations $\geq 0.78$. In contrast, CALIOP-based AOD yields correlations with MODIS $N_d$ of 0.54-0.62 for the two lower AOD quartiles. Moreover, $\sigma_{BC}$ explains 41% of the spatial variance in MODIS $N_d$, whereas AOD only explains 17%, primarily caused by the lack of spatial covariability in the eastern Pacific. Compared with $\sigma_{BC}$, near-surface $\sigma$ weakly correlates in space with MODIS $N_d$, accounting for a 16% variance. It is concluded that the linear regression calculated from $\ln(N_d)$- $\ln(\sigma_{BC})$ (the standard method for quantifying ACI) is more physically meaningful than that derived from the $N_d$-AOD pair.

## 1 Introduction

The anthropogenic forcing by aerosols remains as one of the most elusive aspects of climate change. Its uncertainty is largely attributed to our lack of understanding of the ways that low clouds, especially marine boundary layer clouds in the subtropics, respond to perturbations in tropospheric aerosols (Boucher et al. 2013). Uncertainty in simulating the aerosol effects on clouds (aerosol indirect effects) is evinced in the large spread of state-of-the-art climate models (e.g. Quaas et al., 2009). In addition, models tend to overestimate the strength of aerosol-cloud interactions (ACI) relative to those derived from satellites (Quaas et al., 2009). Satellite studies of ACI typically rely on vertically integrated aerosol properties, namely aerosol optical depth (AOD) or aerosol index (the product of *AOD* and the Angstrom exponent), retrieved over clear-sky scenes. These observations can then be used to quantify the

ACI in terms of fractional change in cloud microphysics relative to fractional changes in aerosol properties (Feingold et al., 2003):

$$ACI = \frac{\partial \ln(c)}{\partial \ln(AOD)} \; , \tag{1}$$


where "*c*" in eq. (1) is a cloud microphysical parameter such as cloud effective radius ($r_e$), cloud optical thickness, cloud droplet number concentration ($N_d$), or a metric for precipitation rate. Since the increase in $N_d$ caused by an increase in cloud condensation nuclei is the fundamental process that leads to the different aerosol-cloud feedbacks, ACI expressed in terms of c = $N_d$ offers a more direct link to the underlying physics of the aerosol indirect effect.

ACI also allows for a simple way of evaluating models with in-situ and satellite observations (e.g. Quaas et al., 2009). However, modeling work by Stier (2016) suggests that the usability of AOD and aerosol index might be limited as a CCN proxy given the inability of a vertically integrated quantity to resolve the aerosol properties in the boundary layer, where the aerosol-droplet activation typically occurs. The importance of counting on vertically resolved observations was further addressed by Shinozuka et al. (2015), who found better correlations between concurrent

airborne 0.55-μm dry aerosol extinction coefficient and CCN than that for the AOD-CCN pair. In addition to issues attributed to the physical representation of AOD and AI, their applicability to ACI studies is hindered by retrieval uncertainties attributed to plausible clear-sky contamination, 3D radiative transfer effects, and aerosol swelling near the cloud edges (e.g. Christensen et al., 2018; Varnai and Marshak, 2018). In sum, the suitability of using satellite-based ACI calculations to evaluate climate models remains uncertain.

Unlike passive satellite sensors, the Cloud-Aerosol Lidar with Orthogonal Polarization (CALIOP) on CALIPSO was designed to retrieve aerosol properties with an unprecedented high vertical resolution (Winker et al., 2010). This allows better isolation of the aerosols situated near the cloud layer, and thus more likely to interact with the cloud, from the rest of the atmospheric column. Moreover, CALIOP's better cloud screening, with aerosols retrievals insensitive to 3D radiative transfer effects, should result in improved aerosol indirect effect estimates

relative to those determined from passive sensors measurements alone. Earlier studies over the southeast Atlantic demonstrated the value of combining the aerosol layer detection capability of CALIOP with passive sensor data from other A-Train satellites (e.g. Costantino and Breon, 2013; Painemal et al. 2014). To our knowledge, the quantitative use of vertically resolved CALIOP aerosol extinction coefficient for ACI studies is nearly non-existent in the literature. In this study, we evaluate the use of CALIOP aerosol extinction coefficient for ACI identification.

Specifically, we analyze the benefit of using near-surface, within-cloud, and above-cloud aerosol extinction coefficient relative to the standard AOD approach. As our interest is to evaluate the value of exploiting the vertically resolved information, we use aerosol extinction coefficient retrievals constrained with a CALIOP-based AOD estimated using the Synergized Optical Depth of Aerosols algorithm (SODA, Josset et al., 2008, 2015; Painemal et al., 2019). This approach is selected because: a) it enables us to compare vertically resolved CALIOP retrievals

against an AOD product (SODA) that is more consistent with other remote sensing and satellite AODs than the standard CALIOP product (Painemal et al., 2019), and b) by using both aerosol extinction coefficient and AOD

derived from CALIOP, we can focus on the physical interpretation of the results rather than disentangling instruments and algorithm differences when using two dissimilar aerosol products. Lastly, low-cloud droplet number concentration ($N_d$) derived from Aqua-MODIS (Grosvenor et al., 2018) is used to quantify aerosol-cloud covariability using different aerosol proxies.

## 2 Methods

### 2.1 Dataset

Daytime aerosol retrievals are taken from a new aerosol extinction coefficient ($\sigma$) dataset derived from CALIPSO attenuated backscatter and SODA AOD (Painemal et al., 2019). SODA algorithm estimates AOD by combining surface ocean return from CALIPSO the CloudSat's Cloud Profiling Radar, and thus, no additional assumptions about the aerosol type and optical properties are required (Josset et al., 2015). In short, $\sigma$ is estimated from the lidar equation constrained with SODA AOD by applying the Fernald-Klett iterative algorithm (Fernald, 1984). The CALIOP-S retrievals employed here are estimated at 1-km along-track resolution assuming a constant extinction-to-backscatter ratio (lidar ratio) with height, and cloudless 1-km pixels (according to CALIPSO Vertical Feature Mask version 4) are used in this study. The dataset is self-consistent in the sense that the vertically integrated extinction coefficient is equivalent to CALIOP-SODA AOD. In addition, using the CALIOP-SODA extinction yields excellent agreement with airborne high spectral resolution lidar retrievals, with SODA AOD showing a better agreement with MODIS AOD than the standard CALIOP version 4 (v4) product (Painemal et al., 2019). We use CALIOP-S instead of the standard CALIOP v4 product because CALIOP-S does not depend on an aerosol classification scheme, enabling retrievals in occasions when CALIOP v4 does not retrieve properties due to the impossibility of classifying tenuous aerosol layers. CALIOP-S is also advantageous as its 1-km along-track resolution is consistent with that of MODIS, whereas CALIPSO v4 aerosol retrievals are estimated at 5-km, 20-km, or 80-km spatial resolution depending on the strength of the aerosol signal. While CALIOP-S agrees better with in-situ data than CALIOP v4 over the tropical western Atlantic (Painemal et al. 2019), it is expected that CALIOP v4 would yield results similar to those presented here, given the high correlation between both datasets; however, this will require further analysis beyond the scope of this paper.

Daytime Aqua-MODIS retrievals of cloud effective radius ($r_e$) and cloud optical depth ($\tau$) at 1-km resolution (at nadir) are estimated using the Clouds and Earth's Radiant Energy System (CERES) Edition 4.0 algorithms (Minnis et al., 2011 and 2020). CERES-MODIS $r_e$ and $\tau$ are estimated, respectively, from MODIS 0.64 and 3.79 μm bands, with the latter being less sensitive to three-dimensional radiative effects and biases due to subpixel inhomogeneity than the more widely used 2.1-μm $r_e$ (Zhang and Platnick, 2011; Painemal et al., 2013). The cloud microphysical variable used in this study is the cloud droplet number concentration $N_d$, which is estimated using the adiabatic formulation (Painemal and Zuidema, 2011; Grosvenor et al., 2018) as:

$$N_d = \Gamma^{1/2} \frac{10^{1/2}}{4\pi \rho_w^{1/2} k} \frac{\tau^{1/2}}{r_e^{5/2}} \tag{2}$$

where $\rho_w$ is the liquid water density and k is cubic ratio between the volume mean radius and the effective radius, assumed constant at 0.8 (Martin et al., 1994). The adiabatic lapse rate of condensation ($\Gamma$, Albrecht et al., 2000) was calculated using CERES MODIS cloud-top temperature and pressure. The reader is referred to Grosvenor et al. (2018) for a thorough review on the satellite-based $N_d$ formulation in eq. (2). For this study, we used 9 months of available collocated CALIOP and MODIS retrievals: April, May, and September 2010, and May to October 2013. The use of boreal spring and summer months should help reduce any role of annual cycle in explaining MODIS/CALIOP correlations.

## 2.2 Matching method

The first step in the analysis is to develop a method for combining cloud and aerosol retrievals, which is summarized in Fig. 1 and described as follows. For each 1-km CALIOP-SODA retrieval, we select 10 MODIS pixels east and 10 pixels west of the CALIPSO ground-track. This configuration considers the fact that aerosol and cloud retrievals cannot be simultaneously retrieved over the same pixel. We further reduce the dataset complexity by averaging CALIOP-S retrievals to achieve a 5-km resolution along the CALIPSO track. Similarly, we average MODIS retrievals every 5x5 pixels (km). That is, for each 5-km CALIOP-S retrieval, two contiguous 5x5km averaged MODIS grids are collocated east and west of the 5-km CALIOP-S point (Figure 1a). Lastly, the collocated aerosol-cloud pair is defined as the 5-km CALIOP-S and 5x5km MODIS retrievals (two grids east and west from CALIOP-S) averaged over 25-km along-track segments (domain of Fig. 1a).

The calculation of near-surface, below cloud-top, and above cloud-top aerosol extinction coefficient is summarized in Fig. 1b. First, cloud top height identification is achieved by means of the CALIOP 333-m cloud layer product (333 m and 30 m, horizontal and vertical resolution, respectively) spatially averaged to 25-km resolution (along-track). Next, below and above cloud top σs are independently calculated as the 300-m averaged layers 60-m below and above cloud top height. Near-surface σ is estimated for a 300-m depth layer located at 60 m above the surface. The 60-m gap is intended to minimize possible uncertainties in the cloud top identification and surface contamination.

## 2.3 Constraining the influence of cloud cover

The extended practice of matching cloudy (cloud retrievals) and neighboring cloudless (aerosol retrievals) pixels for climate research applications raises the question of whether artifacts in aerosol and cloud retrievals, especially in highly inhomogeneous partially cloudy regions, can inadvertently introduce biases in the ACI calculations. While ways for reducing uncertainties in passive AOD for pixels in the vicinity of clouds have been proposed recently (e.g. Christensen et al., 2017), methods applicable to satellite-based ACI studies are lacking. The objective of this section is to describe a method intended to minimize aerosol-cloud correlations primarily modulated by cloud cover via mechanisms not related to microphysical interactions, and possibly associated with retrieval biases. Using the configuration in Fig. 1a, we first attempt to reduce cloud retrieval uncertainties near the cloud edges by

limiting the analysis to 25-km averaged samples constructed from MODIS 5x5 km$^2$ $N_d$ (Fig. 1a) with 5x5 km$^2$ cloud fraction exceeding 0.9 (90%). The comparison of all-sky (no cloud fraction screening) and cloudy $N_d$ (with the CF screening applied to 5x5km$^2$ grids) as a function of 25-km MODIS cloud fraction (Fig. 2) reveals that all-sky $N_d$ dramatically increases with cloud fraction CF (black line) from 15 cm$^{-3}$ to 90 cm$^{-3}$, but a weaker dependence is observed for cloudy $N_d$ (red line). This is consistent with an expected negative bias in $N_d$ (eq. 2) for scenes with reduced cloud cover caused by a positive bias in $r_e$ and negative bias in τ relative to τ and $r_e$ retrievals in spatially homogeneous scenes (Painemal et al., 2013).

In terms of AOD, we found that 25-km CALIOP-S AOD increases with the 25-km MODIS CF at a rather low rate for CF < 0.95 with an increment of less than 0.04, and a rapid increase thereafter to reach a maximum of 0.21, similar to the results in Loeb and Manalo-Smith (2005). In addition, changes in CALIOP-S AOD vary with CALIOP CF in a much narrower range (from 0.10 to 0.12, not shown). The AOD increase for high values of MODIS CF is interpreted as the enhanced swelling of aerosols embedded in regions with extensive cloud cover (Varnai and Marshak, 2018). Fig. 2b depicts how AOD and $N_d$ covary when the data are binned as a function of CF ($|_{CF}$), for 50 cloud fraction bins containing identical number of samples. All-sky $N_d$ increases with AOD (black circles) over a $N_d$ range of almost 80 cm$^{-3}$. Interestingly, the AOD-$N_d$ scatterplot for cloudy $N_d$ and AOD for scenes with MODIS CF < 0.95 (red circles) yields a weaker dependence, with a $N_d$ magnitude of 20 cm$^{-3}$ (after ignoring two outliers). Based on this analysis, we minimize the potential modulation of cloud cover by only using MODIS cloud retrievals with MODIS CF > 0.9 over a 5x5km$^2$ grid and aerosol retrievals embedded in 25-km segments with MODIS CF < 0.95.

## 3 Results

### 3.1 AOD and vertical structure

We first show the median CALIOP-SODA (CALIOP-S) aerosol extinction profiles for four ranges of cloud top height (CTH), which is in turn used as a surrogate for the boundary layer height (Fig. 3a). As expected, σ decreases with increasing height, with a more pronounced reduction in the free troposphere. It is worth noting that even though the mean AOD for each profile is approximately 0.10, σ for the layer below the cloud top varies substantially. For instance, the profile for the shallowest boundary layer (black) yields σ between 0.05-0.08 km$^{-1}$ below cloud top, whereas σ is less than 0.01 km$^{-1}$ near the cloud layer for high clouds with CTH >2.2 km (magenta). The relationship between below cloud-top aerosol extinction coefficient ($\sigma_{BC}$) and AOD is depicted in detail in Fig. 3b. AOD changes with $\sigma_{BC}$ are well characterized by a linear fit for AOD>0.1; however, for $\sigma_{BC}$ (AOD) less than 0.1 km$^{-1}$ (0.1), AOD changes slowly with $\sigma_{BC}$, as determined by the AOD-$\sigma_{BC}$ slope. This gradual gradient is mainly attributed to the scattered AOD-$\sigma_{BC}$ relationship (r = 0.27) for $\sigma_{BC}$ <0.1 km$^{-1}$, suggesting that AOD poorly represents the aerosol optical properties in the boundary layer aerosols for relatively pristine environments. While it has been documented that biases in ACI can be caused by instrument detectability limitations in pristine environments (Ma et al., 2018), the results here suggest that the AOD weak co-variability with boundary layer aerosols is also an important factor that needs to be taken into account.   The weak correlation for $\sigma_{BC}$ <0.1 km$^{-1}$ gives rise to an apparent non-linear

variation of AOD with $\sigma_{BC}$. Overall, these results anticipate a dissimilar dependence of $N_d$ on AOD and $\sigma_{BC}$ as it is demonstrated in the following analysis.

**3.2 Dependence of MODIS $N_d$ on different CALIOP-S aerosol proxies**

The dependencies of MODIS $N_d$ on different CALIOP-S aerosol variables are summarized in Fig. 3c and d, with $N_d$ binned as a function of a CALIOP-S retrieval using an equal frequency binning. MODIS $N_d$ exhibits a nearly linear relationship with CALIOP-S $\sigma_{BC}$ especially for $N_d > 40$ cm$^{-3}$ (Fig 3c, black circles), whereas MODIS $N_d$ varies

linearly with AOD for the 0.1-0.3 range, and shows an anticorrelation for AOD > 0.3. Opposite sign correlations between satellite AOD (AI) and cloud effective radius for AOD>0.2 have also been observed by Bréon et al. (2002). For AOD < 0.1, $N_d$ shows little change with AOD, in agreement with the modest sensitivity of AOD to variations in $\sigma_{BC}$ depicted in Fig. 3b. We also analyze near-surface $\sigma$ ($\sigma_{SFC}$) as a way to assess ground-based observations for investigating ACI (e.g. McComiskey et al., 2009; Liu and Li, 2018). In general, it is found a narrower range of binned

$N_d$ values as a function of $\sigma_{SFC}$ than that for $\sigma_{BC}$ (Fig 3c, red dots). To further confirm the reduced covariability between $N_d$ and AOD and $\sigma_{SFC}$ relative to $\sigma_{BC}$, we computed the standard linear correlation coefficient ($r$) and found that $N_d$-AOD $r$ for the two lowest AOD quartiles (AOD $\leq$ 0.092) are 0.62 and 0.54, whereas $N_d$-$\sigma_{BC}$ $r$ further decreases for $\sigma_{SFC}$ to statistically insignificant values for the lower quartiles (0.32 and 0.06). In contrast, $\sigma_{BC}$ yields the highest $r$ with values of 0.78 and 0.90 ($\sigma_{BC} \leq$ 0.058 km$^{-1}$). Lastly, the monotonic increase of $N_d$ with the $\sigma_{BC}$ indicates that

the AOD-$N_d$ anticorrelation observed for AOD>0.3 in Fig. 3d is not indicative of aerosol-cloud microphysical processes.

Mean MODIS $N_d$ and aerosol fields gridded every 4°x4° are depicted in Fig. 4. The MODIS $N_d$ features a spatial pattern documented in other studies (Fig. 4a, e.g. *Bennartz and Rausch*, 2017) and characterized by high values over the eastern Pacific, eastern Atlantic, and northwestern Pacific. The corresponding CALIOP-S AOD map (Fig.

4b) depicts maxima off the west coasts of Africa and over the Arabian Sea, associated with dust and biomass burning aerosols (Kaufman et al., 2005; Omar et al., 2009; Jickells et al., 2005), whereas AOD remains below 0.1 for vast subtropical areas. As a consistency check, we show in Fig. 5 the map of MODIS AOD at 0.55-μm for the period of study, which agrees well with its CALIOP-S counterpart. The spatial $r$ between the MODIS $N_d$ and CALIOP-S AOD maps is 0.41 (17% explained variance), primarily contributed by the covariability between AOD and $N_d$ over the

southeast Atlantic. Unlike AOD, CALIOP-S $\sigma_{BC}$ produces a local maximum over the northeast Pacific, and a subtle increase along a narrow coastal band off the west coast of South America, consistent with westward gradients in MODIS $N_d$ (Fig.4 c). In addition, the region with high $\sigma_{BC}$ over the southeast Atlantic extends further south relative to that for AOD, in better agreement with MODIS $N_d$. The improved spatial consistency between CALIOP-S $\sigma_{BC}$ and MODIS $N_d$ leads to $r = 0.64$, equivalent to an explained variance of 41%. $\sigma_{SFC}$ spatially correlates at $r = 0.40$, similar

to its AOD counterpart (Fig. 4b, color). Lastly, above-cloud $\sigma$ (contours), with maxima over the eastern Atlantic, has a negligible spatial correlation with MODIS $N_d$ ($r = 0.05$), showing the lack of physical link between free tropospheric aerosols and $N_d$. Spatial co-variability between aerosol concentration and $N_d$ has been verified over the eastern Pacific

with in-situ observations, and manifested in a concurrent westward decrease in CCN and $N_d$ (e.g. Bretherton et al., 2010, Painemal et al. 2015, Bretherton et al., 2019), in agreement with CALIOP-S $\sigma_{BC}$ and MODIS $N_d$ documented here. Although these spatial correlations are promising, such analysis is an oversimplification, as the processes that determine the $N_d$ budget are highly complex and dependent on multiple factors that cannot be accounted for satellite observations only. It is, nevertheless, the goal of this study to explore ways of improving the characterization of aerosol-cloud interactions within the limitations of the current satellite observations.

### 3.3 Aerosol-cloud interaction over the eastern Pacific and Atlantic

We took a closer look at the eastern Pacific and Atlantic, given their dominant albedo susceptibility to changes in cloud microphysics (Painemal 2018). Eastern Pacific/Atlantic aerosol-cloud relationships (Fig. 6a and b) are similar to those in Fig. 3 in that $N_d$ variations with $\sigma_{BC}$ are more linear than the relationship between $N_d$ and AOD. The weak $N_d$-AOD correlation for low AOD, particularly for AOD < 0.1, is evinced again, with $r$ at 0.51 and 0.55 for the Eastern Atlantic and Pacific, respectively (Fig. 6b). In contrast, $r$ for $\sigma_{BC}$ exceeds 0.90 for AOD<0.1 and a simple linear regression captures the aerosol-cloud dependence (Fig. 6a, blue lines). We test the robustness of the aerosol-cloud correlations by minimizing the covariability driven by drizzle (Wood et al., 2012). For this purpose, samples with drizzle rate of more than 1 mm/day, according to the parameterization in Comstock et al. (2004), are removed from the analysis. This threshold reduces the mean precipitation rate from an estimated 2.26 mm/day to 0.28 mm/day (light drizzle according to Wood 2012). The drizzle removal leads to an overall increase in MODIS $N_d$, and more so over the E. Pacific, yielding nearly identical $N_d$ variations with CALIOP-S retrievals between the Pacific and Atlantic regimes (Fig.6c and d). In addition, the $N_d$-$\sigma_{BC}$ correlation remains high ($r \geq 0.80$) for AOD < 0.1, whereas $N_d$-AOD $r$ is less than 0.38 for AOD < 0.1. These results suggest that drizzle strengthens the $N_d$-aerosol relationship, however, aerosols still modulate $N_d$ after moderate and heavy precipitation is removed from the analysis.

### 4 Discussion

While CCN-AOD relationships have been inferred from ground-based observations, the statistics primarily relate near-surface CCN with remote sensing quantities under the specific environmental conditions of the ground-based sites. These limitations are circumvented in Stier (2016) by evaluating cloud-base CCN and AOD simulated by a global climate model. Stier (2016) found that the simulated AOD explains 25% of the cloud-base CCN variance over most of the globe. In this regard, our analysis provides the first extensive observational evidence of Stier (2016) over the subtropical ocean in that AOD explains a modest variance of the aerosol optical properties near the cloud layer in the boundary layer. While we used $N_d$ instead of CCN, this study also reveals that CALIOP-SODA AOD explains a spatial variance of less than 20% over the region where Stier (2016) reported weak CCN-AOD correlations. Even though we found the highest spatial correlation between MODIS $N_d$, and the vertically resolved aerosol extinction coefficient at the cloud level, close attention needs to be paid to the relationship between CCN and $\sigma_{BC}$ (e.g. Shinozuka et al., 2015), which depends on aerosol type, hygroscopicity, and aerosol size distribution, among other factors. For instance, the lack of spatial covariation in the 0-10°N band over the Atlantic Ocean is caused by

the dominant presence of dust, which is a weakly hygroscopic species (Koehler et al., 2009). Moreover, accounting for the aerosol humidification factor in optical retrievals (Gasso et al., 2000) will be key to more closely linking aerosol extinction to CCN. CALIOP-S aerosol index (AI) was not calculated due to uncertainties in the 1064-nm retrievals. However, as AI strongly depends on AOD, similar issues should be expected for AI as well. Indeed, ACI statistics estimated from AI are typically similar to those for AOD (Chen et al., 2014). Moreover, Aqua-MODIS AI in Figure 7 (the product between daytime Level 3 Angstrom exponent and AOD at 550 nm) for the same period of study reveals a spatial pattern poorly correlated with $N_d$, especially over the eastern Pacific, despite an AI decrease over the tropical north Atlantic, where dust is the main aerosol species.

Concerning AOD, two aspects call into question its adequacy for ACI studies: a) the reduced spatial correlation with $N_d$ especially over the eastern Pacific, and b) the non-linear variations of $N_d$ with AOD, caused by the modest changes of $N_d$ for AOD<0.1, further suggesting that the ACI estimation via the AOD-$N_d$ linear fit is less physically meaningful than commonly thought. The lack of linearity and low AOD-$N_d$ correlation for AOD<0.1 is associated with a reduced ability of AOD to capture aerosol variability in the boundary layer (Fig. 3b). Thus, the apparently weak sensitivity of $N_d$ to AOD for pristine environments (Gryspeerdt et al., 2016) is a remote sensing artifact rather than a real cloud microphysical behavior. For more polluted conditions, AOD correlates better with $N_d$, implying that AOD could still be a useful CCN proxy for specific polluted conditions. Similarly, using near-surface $\sigma$ does not offer a significant improvement, at least in the subtropics. This possibly reflects the reduced covariation between near-surface aerosol concentration and that below cloud base in decoupled atmospheric boundary layers (Painemal et al., 2017), in addition to the plausible $\sigma$ enhancement contributed by large sea salt particles near the surface. Given the limited dataset used in this study, the computation of statistically robust ACI maps is left for future work as this will require the use of the full CALIOP data record. However, the consequences of using AOD are evident from Fig. 6a and b for the low aerosol values. Fractional $N_d$ variations (relative to the mean) between the median and the lowest aerosol bins (Fig. 6) are 9.0 % (Eastern Pacific) and 7.8 % (Eastern Atlantic). In contrast, the equivalent $N_d$ fractional changes for $\sigma_{BC}$ are between 31.3-31.54 %, for the below-the-median aerosol range. That is, the computed $N_d$ fractional change as a function of $\sigma_{BC}$ is more than three times greater than that for AOD. Keeping in mind that changes in below-cloud aerosol extinction coefficient are associated with even smaller AOD variations (e.g. Fig. 3a and b), this simple susceptibility calculation suggests that computed changes in both $N_d$ and shortwave fluxes (see eq. 3 in Gryspeerdt et al., 2017) due to changes in AOD would substantially underestimate the actual ACI radiative forcing for pristine conditions.

Another key aspect for reducing uncertainties in satellite estimates is to develop methods that can minimize biases in partially cloudy scenes. For instance, when the spatial correlation analysis in Fig. 4 was repeated with unscreened cloud and aerosol retrievals (not shown), spatial $r$ for $N_d$-AOD and $N_d$-$\sigma_{BC}$ were weaker (0.26 and 0.48) than those estimated from the screened dataset (0.41 and 0.61). The analysis also suggests that the use of level 3 products, with 1˚ spatial resolution, for estimating aerosol-cloud interaction relationships will likely be biased due to the additive effect of sub-pixel variability and 3D radiative effects in cloud retrievals, and aerosol swelling in regions with high cloud cover.


**5 Concluding remarks**

Readily available AOD from ground-based sensors and satellites have become the most commonly used variable for quantifying aerosol-cloud interactions. Despite a general positive correlation between AOD and CCN (Andreae 2009) for a broad range of pollution conditions, the relationship is weaker over the ocean (Stier et al., 2016).

Here, we show for the first time that CALIOP-based aerosol extinction coefficient retrievals are central to reducing uncertainties in observationally-based estimates of aerosol indirect effects. Below-cloud-top CALIOP-S aerosol extinction is the retrieval that yields the strongest correlation with MODIS $N_d$ compared with: AOD, near-surface, and above cloud-top aerosol extinction coefficient. The fact that the $\log(N_d)$-$\log$(CCN) relationship over the ocean is also linear when estimated from in-situ observations (Twohy et al., 2005; Painemal and Zuidema, 2013; Painemal et

al., 2017) lends confidence in the results presented here. While our analysis shows that precipitation strengthens the dependence of $N_d$ on $\sigma_{BC}$, it also reveals that the $\sigma_{BC}$ -$N_d$ correlation remains high for non-precipitating/light-drizzling samples. We note that precipitation retrievals from CloudSat have not been utilized here, and thus, the full evaluation of precipitation susceptibility is left for future work. However, since several satellite-based ACI metrics also rely on AOD (Sorooshian et al., 2010), the results of our study should be applicable to other aspects of the aerosol indirect

effect quantified from satellite observations. Given the extensive use of AOD-based results for evaluating ACI in climate models, caution needs to be exercised before interpreting linear regressions between AOD and cloud microphysics as meaningful quantification of the cloud response to aerosols. In light of the results presented here, it would be informative to assess the extent of which aerosol extinction coefficient can be combined with climate models to quantify the ACI radiative forcing since pre-industrial conditions as in Gryspeerdt et al. (2017) but

expressing ACI in terms of $\sigma_{BC}$ instead of MODIS AI. Unfortunately, given the remaining challenges in simulating the aerosol vertical structure in climate models (e.g. Koffi et al., 2016), it is uncertain that the simulated relationships between CCN, AOD, aerosol extinction, and vertical variability can be used to analyze the advantages and disadvantages of using a specific CCN proxy.

Moving beyond the AOD paradigm is crucial to providing a trustworthy benchmark that can be applied to

the evaluation of climate models. As demonstrated here, CALIOP is central in advancing toward this goal; and thus, future efforts should be oriented to exploit the nearly 12 years of CALIPSO measurements collocated with the A-Train satellite constellation. Lastly, coordinated field campaigns that explore the link between CCN, aerosol extinction coefficient, and cloud microphysics using in-situ and remotely-sensed observations will be essential to develop new approaches for estimating ACI and quantifying uncertainties in satellite-based assessments. A promising

strategy has been adopted by Aerosol meteorology Interactions oVer the western ATlantic Experiment (ACTIVATE). ACTIVATE deployment of two airplanes flying in formation will allow for the collocation of in-situ and remotely sensed aerosol and cloud properties with unprecedented spatial resolution (Sorooshian et al., 2019). The proper characterization of aerosols using diverse sensors will be fundamental for helping reconcile different ACI estimates and provide a more physically reliable benchmark for the evaluation of climate models.


**Acknowledgements**

This work was funded by the CloudSat and CALIPSO Science Recompete Program NASA award # NNH16CY04C. W.S. and P.M. acknowledge the support of the CERES program.

**Author contributions**

D.P. designed the study and wrote the manuscript with contributions of all the co-authors. F-L.C created the matched MODIS-CALIOP-SODA dataset and M.C produced the SODA-CALIOP aerosol retrievals.

**Competing interests**

The authors declare no competing interests.

**Data availability**

The SODA aerosol optical depth is developed at the AERIS/ICARE data and services center (http://www.icare.univ-lille1.fr/projects/soda) in Lille (France) in the frame of the CALIPSO mission and supported by CNES. CALIPSO
v4 cloud products are available at https://www-calipso.larc.nasa.gov, and CERES-MODIS cloud products at https://ceres.larc.nasa.gov/order_data.php. CALIOP-SODA aerosol extinction coefficients and lidar ratios are available via sftp at: calipso_soda@xfr999.larc.nasa.gov (further instructions provided upon request).

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

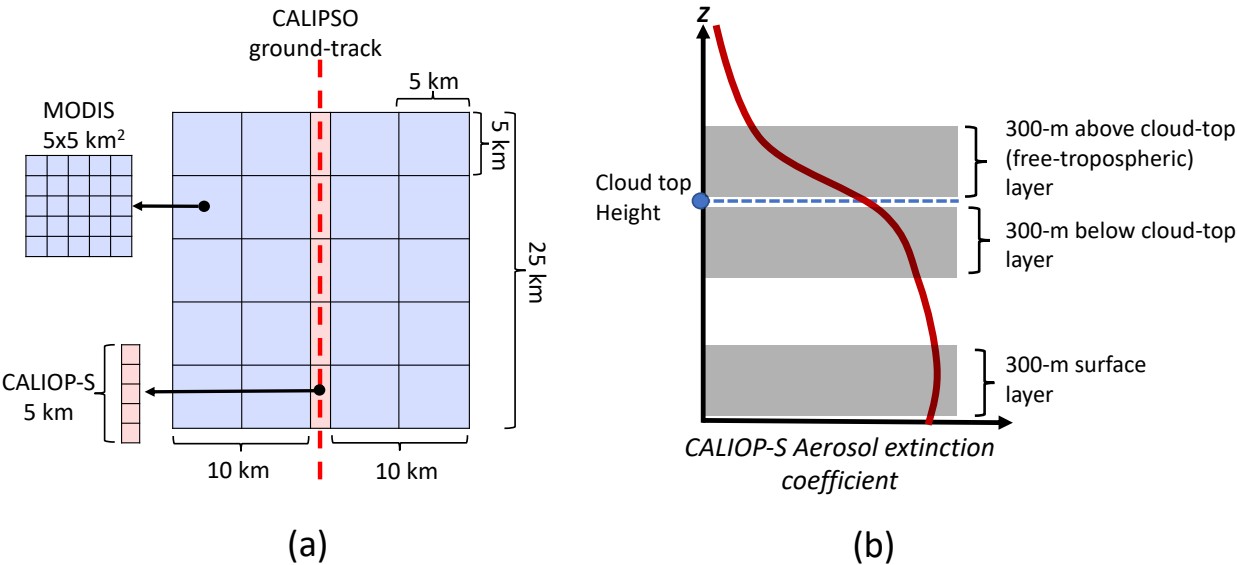

*Figure 1: A) Horizontal view of the matching configuration along a 25-km CALIPSO track. Light blue indicates MODIS cloud retrievals, and light red CALIOP-S pixels. B) Idealized aerosol extinction profile (red) and the location of the 300-m layers used to calculate near-surface, below cloud-top, and above cloud-top aerosol extinction coefficient.*

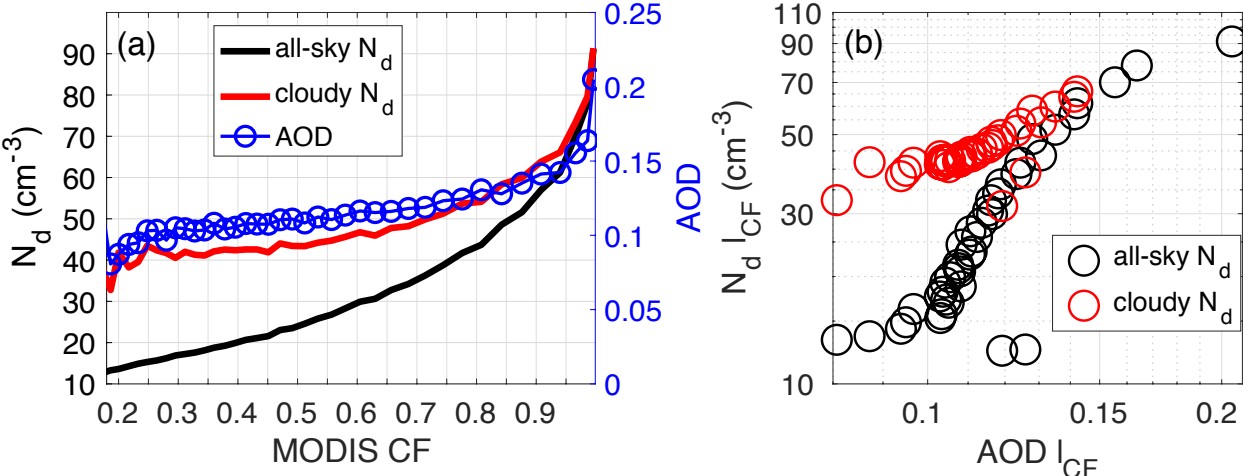

*Figure 2: a) 25-km CALIOP-S AOD and MODIS $N_d$ binned as a function of 25-km MODIS cloud fraction. Cloudy $N_d$ refers to 25-km $N_d$ calculated using 5x5 km$^2$ blocks (Fig. 1a) with cloud fraction exceeding 0.9. b) Variation of CALIOP-S AOD and MODIS $N_d$ binned as a function of MODIS CF ($|_{CF}$) from Fig. S4a. Red circles represent cloudy $N_d$ and CALIOP-S AOD with 25-km MODIS CF < 0.95. Black circles represent binned $N_d$ and AOD irrespective of their 5x5 km$^2$ and 5km cloud fraction, respectively.*

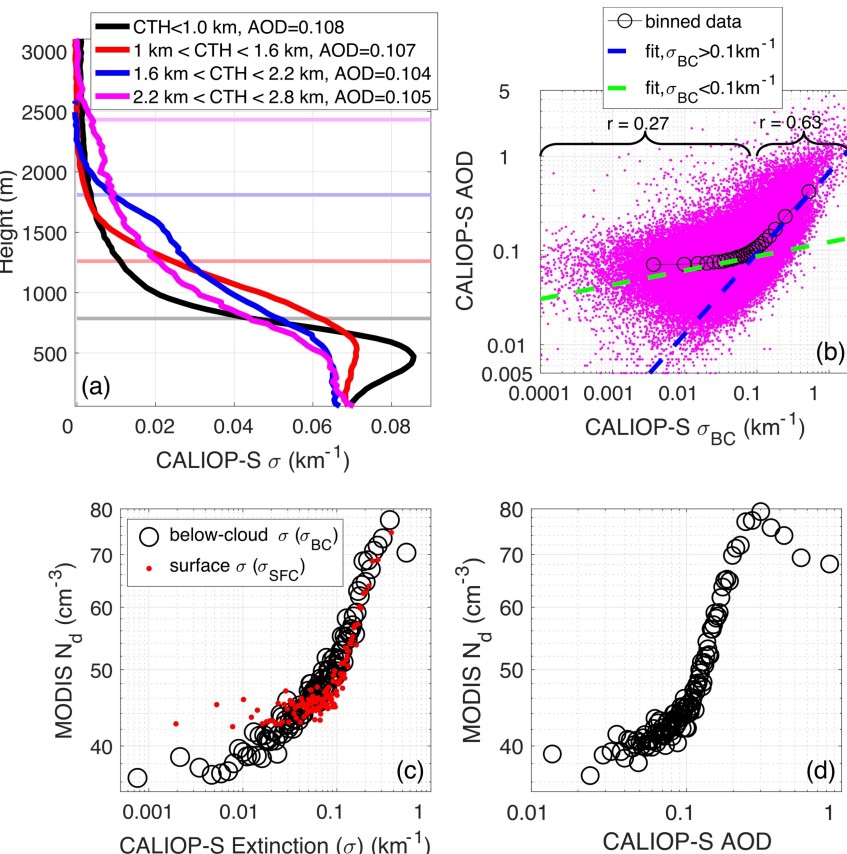

*Figure 3: a) Median vertical profiles of aerosol extinction coefficient for different cloud top heights (CTH), with the horizontal lines (light color) representing the median CTH for each group. b) Relationship between AOD and $\sigma_{BC}$, binned data (black circle), and linear correlation coefficients and regressions for CALIOP-S $\sigma_{BC}$ greater and lower than 0.1 km$^{-1}$ (dashed blue and green lines, respectively). Mean MODIS $N_d$ binned as a function of: c) $\sigma_{BC}$ (black circles) and $\sigma_{SFC}$ (red dots) and d) AOD. Data are taken over the subtropical ocean (35˚S-35˚N).*

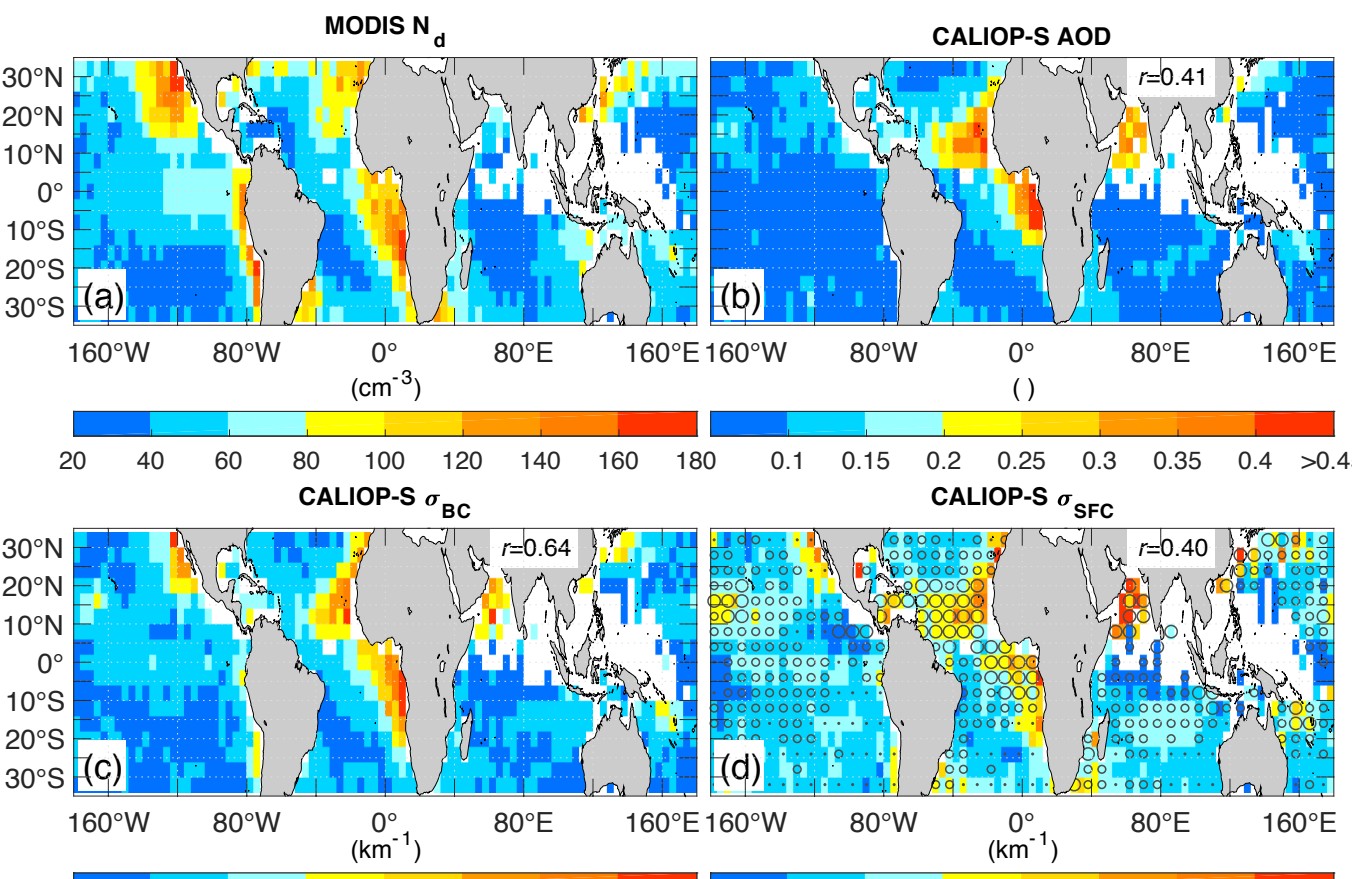

*Figure 4: 4˚x4˚ maps derived from 9 months of data (Methods): a) MODIS $N_d$, b) CALIOP-S AOD, c) $\sigma_{BC}$, and d) $\sigma_{SFC}$ (colors) and above-cloud aerosol extinctions (symbols: dots, small, and large circles denotes the following ranges: 0-0.015, 0.015-0.03, and 0.03-0.45 $km^{-1}$). White areas correspond to regions with less than 25 samples (10% of the observed maximum). The spatial linear correlation coefficient (r) between MODIS $N_d$ and the specific CALIOP-S map is also included.*






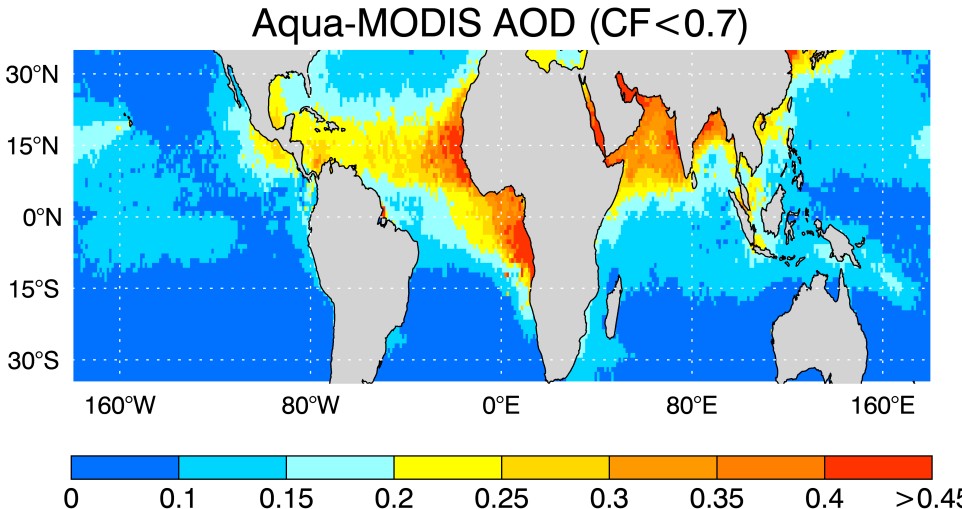

*Figure 5: Mean Aqua-MODIS 0.55-μm AOD for the period of study based on MODIS Collection 6 Level 3 product (MYD08_D3). Potential aerosol biases and swelling are reduced by removing 1˚ grids with cloud fraction (Collection 6) exceeding 0.7. The spatial pattern is nearly identical for the map without filtering (not shown).*

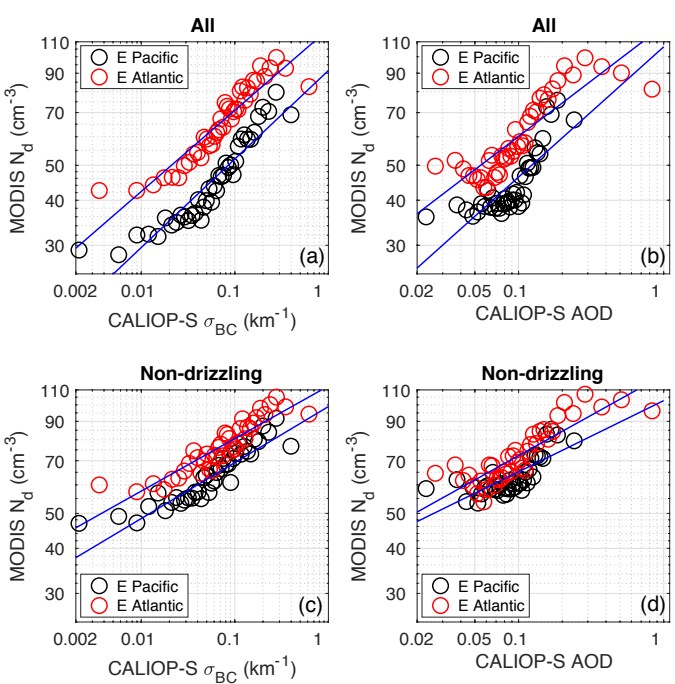

*Figure 6: MODIS $N_d$ binned as a function of A) $\sigma_{BC}$ and B) AOD over the eastern (E) Pacific (150˚W-110˚W, 20˚N-35˚N; 110˚W-70˚W, 10˚S-30˚S) and Atlantic (50˚W-15˚W, 20˚N-35˚N; 15˚W-15˚E, 0˚-30˚S) regions, defined by the boxes in Fig. 1. C) and D) as in A) and B) after removing drizzling samples. Linear regression (logarithmic scale) are depicted in blue.*


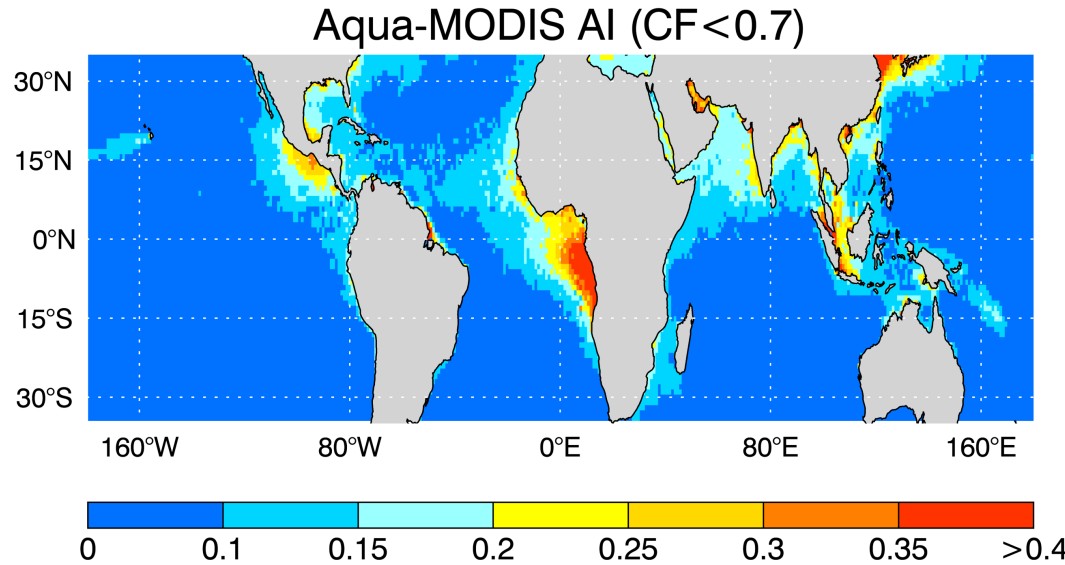

*Figure 7: As Fig. 5 but for aerosol index derived from Aqua-MODIS AOD and Angstrom exponent.*
