# Peer review of "Reducing uncertainties in satellite estimates of aerosol-cloud interactions over the subtropical ocean by integrating vertically resolved aerosol observations"

_Atmospheric Chemistry and Physics, 2019_

## Referee Comment (RC1) · Anonymous Referee #2 · 27 Nov 2019

Reducing uncertainties in satellite estimates of aerosol-cloud interactions over the subtropical ocean by integrating vertically resolved aerosol observations by Painemal et al. studies the relationship between CALIPSO products of aerosol properties and MODIS-derived cloud droplet number concentration. Stronger correlation is shown when the aerosol properties are resolved for altitude than when they are for the entire column.

The paper is easy to follow. The conclusion is, while not surprising, a useful reminder of a limitation in past satellite-based studies of aerosol-cloud interactions. I recommend publication. I only have minor comments.

[Figure]

Line 114. Insert a space in Fig. 1and.

Line 123. Clarify that 333-m and 25-km refer to horizontal resolution.

Line 198. 4c should read 4b.

Line 204. To emphasize the similarity, unify the axis limits for Nd in Fig. 2, 3 and 6.

Line 216. Are the (mildly) higher r values for the all cases interpreted as strengthening of the relationship by drizzle? Can they be partly explained by the larger number of samples?
* * *

---

## Referee Comment (RC2) · Anonymous Referee #1 · 1 Jan 2020

In this work, the authors investigate the correlation between the cloud droplet number concentration and aerosol properties, demonstrating the important of information on the vertical location of the aerosol. The show that when aerosol close to (but below) the cloud top is used instead of height-integrated aerosol optical depth, the correlation between aerosol and MODIS $N_d$ is improved.

The paper is well written with appropriate figures and illustrates a useful result that had previously been suggested using model data. I have a few overarching comments about the analysis and a couple of smaller comments, but after this, I would be happy

to recommend publication in Atmospheric Chemistry and Physics.

**Major points**

The paper starts off by mentioning the ACI construct, but then doesn't seem to refer to the ACI again. One very interesting potential consequence of this work is the impact on the ACI of the biases identified here. Is the error in the ACI using AOD random (in which case the mean ACI and RFaci are relatively unchanged) or is it a systematic bias, which would affect RFaci quantification. This is not a large addition to the manuscript, but it would be a potentially significant result that might not otherwise stand as a paper on its own.

Secondly, there appears to be a reliance on spatial correlations across the globe. Given that cloud properties vary across the globe and that these variations might be expected to generate variations in the CCN-$N_d$ relationship (e.g. through updraft variations), is this not slightly dangerous? One factor that might be interesting is global maps of the correlations (or ACI), which might help to identify any regional effects.

**Minor comments**

L34 - product of

L45 - Stier (2016) noted that vertical mismatches are important to get a high correlation, but results using other aerosol-climate models (Gryspeerdt et al, PNAS, 2017) suggested that this does not have a large impact on the inferred RFaci (as long as the PD-PI aerosol product is appropriate). This paper focuses on the aerosol-$N_d$ correlation, which is one step further removed from the RFaci. As the initial justification of this work is based around the RFaci, it would be good to mention how these results are linked to it/might affect it.

L117 - CALIPSO with CloudSat's

Section 2.2 - It sounds like the MODIS pixels are only paired with CALIOP pixels across the track, rather than along the track. This would mean that some closer pixels are

ignored, as they might be in the along-track direction.

L142 - Zhang and Platnick (JGR, 2011 - their Fig. 14) suggest that the 3.7um channel re is (slightly) negatively biased in in-homogeneous cases. how does this fit with the reasoning here? Also, is the 'cloudy-sky' retrieval mostly limited to high CF scenes (and might this explain the weaker $N_d$-CF relationship)?

L149 - I am unclear on how this binning procedure works. Not much needs to be included here, but it should be specified. What CF range is used?

L161 - I was probably just being slow, but it was only once I got here that I realized the below and above cloud top $\sigma$ values are separate. Perhaps just make the in L124 a plural ($\sigma$s) to make is clearer?

L168 - This has been noted in previous studies (e.g. Ma et al, Nature Communications, 2018)

L176 - Is this unexpected? Several studies in the past have hinted at a saturation of aerosol effects at high AOD.

L179 - It is stated here that there is a narrower range of $N_d$ for each value of $\sigma_{SFC}$, but that it has a lower correlation coefficient with $N_d$ that $\sigma_{BC}$. How does these observations fit together?

L182 - Give that the relationship is non-linear, would a measure such as Spearman's rank be more appropriate that an correlation coefficient?

L198 - It is difficult to compare the contours with the MODIS $N_d$ image. I am not sure that this needs an extra panel, but I can't see a clear alternative. It could even just be stated I think (as it is not a vital part of this work)

Fig 5 and 7 - Could these be included in Fig. 4? Especially as they are mainly here to compare to 4a

---

## Author Response (AR1)

**acp-2019-892 (Painemal et al.)**

**Responses to Reviewer 1**

We appreciate the reviewer's insightful comments and suggestions. His/her comments and recommendations are addressed below (in blue).

The paper starts off by mentioning the ACI construct, but then doesn't seem to refer to the ACI again. One very interesting potential consequence of this work is the impact on the ACI of the biases identified here. Is the error in the ACI using AOD random (in which case the mean ACI and RFaci are relatively unchanged) or is it a systematic bias, which would affect RFaci quantification. This is not a large addition to the manuscript, but it would be a potentially significant result that might not otherwise stand as a paper on its own.

The reviewer raises an important point. We decided to focus on the implicit assumption of the ACI construct, that is, the linearity of the aerosol-cloud relationship (in logarithmic scale). While we agree with the reviewer that the ACI quantification is relevant, this is left for future work, as robust estimates will require the use of the full CALIPSO data record. However, following the reviewer's suggestions, we have added the following paragraph in Section 4:

"Given the limited dataset used in this study, the computation of statistically robust ACI maps is left for future work as this will require the use of the full CALIOP data record. However, the consequences of using AOD are evident from Fig. 6a and b for the low aerosol values. Fractional $N_d$ variations (relative to the mean) between the median and the lowest aerosol bins (Fig. 6) are 9.0 % (Eastern Pacific) and 7.8 % (Eastern Atlantic). In contrast, the equivalent $N_d$ fractional changes for $\sigma_{BC}$ are between 31.3-31.54 %, for the below-the-median aerosol range. That is, the computed $N_d$ fractional change as a function of $\sigma_{BC}$ is more than three times greater than that for AOD. Keeping in mind that changes in below-cloud aerosol extinction coefficient are associated with even smaller AOD variations (e.g. Fig. 3a and b), this simple susceptibility calculation suggests that computed changes in both $N_d$ and shortwave fluxes (see eq. 3 in Gryspeerdt et al., 2017) due to changes in AOD would substantially underestimate the actual ACI radiative forcing for pristine conditions."

(*note*: Dear Reviewer, this response is an updated version of the one available in the interactive discussion)

Secondly, there appears to be a reliance on spatial correlations across the globe. Given that cloud properties vary across the globe and that these variations might be expected to generate variations in the CCN-$N_d$ relationship (e.g. through updraft variations), is this not slightly dangerous? One factor that might be interesting is global maps of the correlations (or ACI), which might help to identify any regional effects.

We agree with the reviewer that, for instance, updraft velocity is an important factor to be taken into account for understanding the spatial distribution of aerosols that ultimately module the cloud droplet number concentration (e.g. Sullivan et al. 2016). Similarly, precipitation and entrainment are likely contributors to the Nd and CCN budget. However, despite the complexity of the cloud-aerosol microphysical processes, observational studies have shown concomitant westward gradients in aerosol concentration and cloud droplet number concentration over the eastern Pacific and northeast Atlantic (e.g. Bretherton et al., 2010; Painemal et al., 2015; Bretherton et al., 2019; Wood et al., 2019), where low aerosol concentrations are associated with low $N_d$ (e.g. Wood et al., 2019, Painemal et al., 2015). So spatial co-variability between aerosol properties and $N_d$ should be expected in subtropical boundary layer clouds. In the revised manuscript, we caution the reader that the spatial correlation approach is an oversimplification of the processes that control aerosol activation, yet the spatial correlation has been corroborated in in-situ

observations over the eastern Pacific and northeast Atlantic. Regarding this point we included the following sentence in the revised manuscript:

"Spatial co-variability between aerosol concentration and $N_d$ has been verified over the eastern Pacific with in-situ observations, and manifested in a concurrent westward decrease in CCN and $N_d$ (e.g. Bretherton et al., 2010, Painemal et al. 2015, Bretherton et al., 2019), in agreement with CALIOP-S $\sigma_{BC}$ and MODIS $N_d$ documented here. Although these spatial correlations are promising, such analysis is an oversimplification, as the processes that determine the $N_d$ budget are highly complex and dependent on multiple factors that cannot be accounted for satellite observations only. It is, nevertheless, the goal of this study to explore ways of improving the characterization of aerosol-cloud interactions within the limitations of the current satellite observations."

As discussed in our response to Comment 1, the ACI mapping will be included in a standalone manuscript. Regarding regional correlations, they are reported for the eastern Atlantic and Pacific only. Given the sparse CALIOP's spatial coverage, we will need a much larger dataset in order to report regional correlations for the entire globe (which we hope to include in the standalone manuscript).

References:

Sullivan SC, Lee D, Oreopoulos L, Nenes A (2016) Role of updraft velocity in temporal variability of global cloud hydrometeor number. Proc Natl Acad Sci USA 113:5791–5796.

Bretherton, C. S., Wood, R., George, R. C., Leon, D., Allen, G., and Zheng, X.: Southeast Pacific stratocumulus clouds, precipitation and boundary layer structure sampled along 20° S during VOCALS-REx, Atmos. Chem. Phys., 10, 10639–10654, https://doi.org/10.5194/acp-10-10639-2010, 2010.

Bretherton, C.S., I.L. McCoy, J. Mohrmann, R. Wood, V. Ghate, A. Gettelman, C.G. Bardeen, B.A. Albrecht, and P. Zuidema, 2019: Cloud, Aerosol, and Boundary Layer Structure across the Northeast Pacific Stratocumulus–Cumulus Transition as Observed during CSET. *Mon. Wea. Rev.,* **147**, 2083–2103.

Painemal, D., Minnis, P., and Nordeen, M. ( 2015), Aerosol variability, synoptic-scale processes, and their link to the cloud microphysics over the northeast Pacific during MAGIC. *J. Geophys. Res. Atmos.*, 120, 5122– 5139. doi: 10.1002/2015JD023175.

Wood, R., Stemmler, J. D., Rémillard, J., and Jefferson, A.( 2017), Low-CCN concentration air masses over the eastern North Atlantic: Seasonality, meteorology, and drivers, *J. Geophys. Res. Atmos.*, 122, 1203– 1223, doi:10.1002/2016JD025557.

L34 - product of
Modified, thanks.

L45 - Stier (2016) noted that vertical mismatches are important to get a high correlation, but results using other aerosol-climate models (Gryspeerdt et al, PNAS, 2017) suggested that this does not have a large impact on the inferred RFaci (as long as the PD-PI aerosol product is appropriate). This paper focuses on the aerosol-$N_d$ correlation, which is one step further removed from the RFaci. As the initial justification of this work is based around the RFaci, it would be good to mention how these results are linked to it/might affect it.

We appreciate the reviewer's suggestions. We have included the following sentence in the revised Discussion:

"In light of the results presented here, it would be informative to assess the extent of which aerosol extinction coefficient can be combined with climate models to quantify the ACI radiative forcing since pre-industrial conditions as in Gryspeerdt et al. (2017) but expressing ACI in terms of $\sigma_{BC}$ instead of MODIS AI"

Going beyond this discussion is difficult as Stier (2016) and Gryspeerdt et al. (2017) rely on climate models for their aerosol assessment. Keeping in mind the limitations of climate models in simulating the aerosol vertical structure (e.g. Koffi et al., 2016), it is unclear whether the simulated relationships between different aerosols proxies are valid representation of those observed in nature. In contrast, our work present an observationally based perspective of Stier (2016). This discussion is included in Section 5:

"Unfortunately, given the remaining challenges in simulating the aerosol vertical structure in climate models (e.g. Koffi et al., 2016), it is uncertain that the simulated relationships between CCN, AOD, aerosol extinction, and vertical variability can be used to analyze the advantages and disadvantages of using a specific CCN proxy."

Reference:
Koffi, B., et al. ( 2016), Evaluation of the aerosol vertical distribution in global aerosol models through comparison against CALIOP measurements: AeroCom phase II results, *J. Geophys. Res. Atmos.*, 121, 7254– 7283, doi:10.1002/2015JD024639.

L117 - CALIPSO with CloudSat's

Section 2.2 - It sounds like the MODIS pixels are only paired with CALIOP pixels across the track, rather than along the track. This would mean that some closer pixels are ignored, as they might be in the along-track direction.
        The reviewer is correct. This configuration was designed for minimizing clear-sky contamination in MODIS, and aerosol swelling near the cloud edges in CALIOP extinctions. By only considering MODIS samples at least 1 km east/west from the CALIOP track, with the filtering discussed in section 2, we are minimizing the potential swelling effect on aerosol extinction by removing aerosol embedded in regions with extensive cloud cover. Recall that simultaneous retrievals of cloud and aerosol in time and space are not possible, and therefore any satellite-based method has to rely on matching neighboring cloud and aerosol pixels.

L142 - Zhang and Platnick (JGR, 2011 - their Fig. 14) suggest that the 3.7um channel re is (slightly) negatively biased in in-homogeneous cases. how does this fit with the reasoning here? Also, is the 'cloudy-sky' retrieval mostly limited to high CF scenes (and might this explain the weaker $N_d$-CF relationship)?
        Zhang and Platnick (2011) and our research (e.g. Painemal et al., 2013) show that 3.7-um cloud effective radius ($r_e$) is smaller than its 2.1-um counterpart; however, this should not interpreted as a bias in 3.7-um $r_e$. Instead, the differences mentioned by the reviewer are the consequence of the higher sensitivity of the 2.1-um channel to spatial inhomogeneity. In contrast, the 3.7-um is less sensitive to sub-pixel inhomogeneity and 3D radiative transfer effect. The advantages of the 3.9-um channel for retrieving $r_e$ were briefly discussed in L96-L98.
        Regarding the second point: cloudy $N_d$ refers to $N_d$ averaged over 5kmx5km grid with cloud fraction>0.9 (90%). These 5kmx5km averaged $N_d$'s are then further averaged in 25-km segments along the CALIPSO track.

L149 - I am unclear on how this binning procedure works. Not much needs to be included here, but it should be specified. What CF range is used?
        CF was averaged using 50 bins with each of them containing the same number of samples. This binning is better described in the revised version.

L161 - I was probably just being slow, but it was only once I got here that I realized the below and above cloud top σ values are separate. Perhaps just make the in L124 a plural (σs) to make is clearer?
We appreciate the reviewer's suggestions. His/her comment was addressed accordingly.

L168 - This has been noted in previous studies (e.g. Ma et al, Nature Communications, 2018)
We appreciate the reviewer's comment. This is indeed a relevant work that is now discussed in the revised Section 3.1.:

"While it has been documented that biases in ACI can be caused by instrument detectability limitations in pristine environments (Ma et al., 2018), the results here suggest that the AOD weak co-variability with boundary layer aerosols is also an important factor that needs to be taken into account."

L176 - Is this unexpected? Several studies in the past have hinted at a saturation of aerosol effects at high AOD.

The reviewer is correct in that aerosol effect saturation at high AOD has been previously observed (e.g. Breon et al., 2002, included in Section 3.2). Nevertheless, our analysis based on $\sigma_{BC}$ suggests the AOD-$N_d$ saturation is possibly an artifact rather than explained by thermodynamical processes described in Feingold et al. (2001).

References:
Bréon, F.-M., Tanré, D., and Generoso, S.: Aerosol effect on cloud droplet size monitored from satellite, Science, 295, 834–838, 2002.
Feingold, G., Remer, L. A., Ramaprasad, J., and Kaufman, Y. J. ( 2001), Analysis of smoke impact on clouds in Brazilian biomass burning regions: An extension of Twomey's approach, *J. Geophys. Res.*, 106( D19), 22907– 22922, doi:10.1029/2001JD000732.

L179 - It is stated here that there is a narrower range of $N_d$ for each value of $\sigma_{SFC}$, but that it has a lower correlation coefficient with $N_d$ that $\sigma_{BC}$. How does these observations fit together?

The sentence refers to the $N_d$ range relative to the range of $\sigma_{SFC}$, rather than individual values of extinction. Therefore, the narrow range of variability in $N_d$ combined with a more scattered relationship give rise to lower correlations with respect to $N_d$-$\sigma_{BC}$. In the revised manuscript, we rephrased the sentence to read:
"In general, it is found a narrower range of the binned $N_d$ as a function $\sigma_{SFC}$ than that for $\sigma_{BC}$."

L182 - Give that the relationship is non-linear, would a measure such as Spearman's rank be more appropriate that an correlation coefficient?

We did calculate the Spearman's rank correlation in a previous version of the manuscript, and the differences were modest, which is why we did not include them in the manuscript. The standard Pearson correlation is more appealing as it allows inferences about variance.

L198 - It is difficult to compare the contours with the MODIS $N_d$ image. I am not sure that this needs an extra panel, but I can't see a clear alternative. It could even just be stated I think (as it is not a vital part of this work)

In the revised the manuscript, we use dots and circles to represent regions with free tropospheric aerosols: 0-0.015, 0.015-0.03, 0.03-0.45 km$^{-1}$. The manuscript was updated with the new figure (see below)

[Figure]

Figure 1: New surface and free tropospheric extinction coefficient in Fig. 4.

Fig 5 and 7 - Could these be included in Fig. 4? Especially as they are mainly here to compare to 4a

While we agree that the inclusion of Fig 5 and 7 into Fig. 4, we prefer to keep them as separate figures, as they are only used to support the results derived from CALIOP-S.

**Responses to Reviewer 2**

We appreciate the reviewer's comments and suggestions. His/her comments and recommendations are addressed below (in blue).

The paper is easy to follow. The conclusion is, while not surprising, a useful reminder of a limitation in past satellite-based studies of aerosol-cloud interactions. I recommend publication. I only have minor comments

We appreciate the reviewer's positive feedback. We agree with the reviewer that uncertainties should be expected when using satellites observations for quantifying aerosol-cloud interactions. However, little has been done to understand the ACI limitations using nearly-global observations, and how the ACI calculation can be improved using vertically resolved aerosol properties, which are two main goals of our manuscript.

Line 114. Insert a space in Fig. 1and.
Line 123. Clarify that 333-m and 25-km refer to horizontal resolution.
Line 198. 4c should read 4b.

We appreciate the reviewers suggestions, which are implemented in the revised version

Line 204. To emphasize the similarity, unify the axis limits for Nd in Fig. 2, 3 and 6.

Since the focus of each figure is to contrast AOD against aerosol extinction coefficient, we decided to keep the axis limits of the original submission.

Line 216. Are the (mildly) higher r values for the all cases interpreted as strengthening of the relationship by drizzle? Can they be partly explained by the larger number of samples?

We interpret the strengthening of the aerosol-cloud relationship by drizzle as the effect of collision/coalescence and droplet removal on the net decrease in $N_d$. This is typically more important in more pristine environments, whereas drizzle is less frequent for more polluted conditions. This produces a change in the aerosol-cloud relationship that is more weighted toward region with lower $N_d$ and aerosol extinction coefficient. Since the $N_d$-extinction relationship is linear, and the scatterplot was constructed using binned values containing the same number of samples, the larger number of samples mentioned by the reviewer should have a small effect.

---

## Author Response (AR2)

Dear Dr. Hatzianastassiou,

We appreciate the positive decision regarding our manuscript. Regarding the following comment:

Actually, AI is not appropriately defined in the manuscript, it first appears in line 51 without being defined, whereas afterwards there seems to be a confusion with it. It is "Aerosol Index", right? In fact, later on, you refer to MODIS AI, and this is also the case of Figure 7 (Aqua-MODIS AI). But, AI is not derived by MODIS, it is an OMI product. Can you please clarify this

We indeed derived the aerosol index from MODIS, as the product between MODIS aerosol optical depth at 550 nm and its corresponding Angstrom exponent retrieval using Level 3 daily data. We did not use OMI aerosol index AI. OMI AI is not defined in the same way as the AI used in the aerosol-cloud interaction community, and therefore its use in the context of our manuscript would be inadequate.
We revised the manuscript to read:

"Moreover, Aqua-MODIS AI in Figure 7 (the product between daytime Level 3 Angstrom exponent and AOD at 550 nm) for the same period of study reveals a spatial pattern poorly correlated with $N_d$, especially over the eastern Pacific, despite an AI decrease over the tropical north Atlantic, where dust is the main aerosol species."
We also updated Figure 7 caption to explain the way AI was calculated.